# N-(n-Butyl) Thiophosphoric Triamide (NBPT)-Coated Urea (NCU) Improved Maize Growth and Nitrogen Use Efficiency (NUE) in Highly Weathered Tropical Soil

**Muhammad Muhaymin Mohd Zuki [1], Noraini Md. Jaafar [1,*], Siti Zaharah Sakimin [2] and Mohd Khanif Yusop [1]**

1   Department of Land Management, Faculty of Agriculture, Universiti Putra Malaysia,
    Serdang 43400, Malaysia; muhammadmuhaymin@ymail.com (M.M.M.Z.); khanif@upm.edu.my (M.K.Y.)
2   Department of Crop Science, Faculty of Agriculture, Universiti Putra Malaysia, Serdang 43400, Malaysia;
    szaharah@upm.edu.my
*   Correspondence: j_noraini@upm.edu.my; Tel.: +03-976-949-53

**Abstract:** Nitrogen (N) fertilizer is commonly used to supply sufficient N for plant uptake, for which urea is one of the highly preferred synthetic N fertilizers due to its high N content. Unfortunately, N provided by urea is rapidly lost upon urea application to soils through ammonia volatilization, leaching, and denitrification. Thus, treatment of urea with urease inhibitor (N-(n-Butyl) Thiophosphoric Triamide (NBPT)) is among the solutions to slow down urea hydrolysis, therefore reducing loss of $NH_3$ and saving N available for plant uptake and growth. A field study was carried out to evaluate the effects of NBPT-coated urea (NCU) at varying rates on growth, yield, and nitrogen use efficiency (NUE) of maize in tropical soil. The experiment was conducted at Field 15, Universiti Putra Malaysia, Serdang, Selangor, Malaysia, and maize (*Zea mays* var. Thai Super Sweet) was used as the test crop. The results showed that all maize grown in soils applied with urea coated with NBPT (NCU) (T2, T3, T4, and T5) had significantly ($P \leq 0.05$) higher chlorophyll content compared to the control (T0 and T1). The surface leaf area of maize grown in NCU-treated soils at 120 kg N h$^{-1}$ (T3) was recorded as the highest. NCU at and 96 kg N ha$^{-1}$ (T3 and T4) were relatively effective in increasing maize plant dry weight, yield, and N uptake. Improvement of NUE by 45% over urea was recorded in the treatment of NCU at 96 kg N ha$^{-1}$. NBPT-coated urea (NCU) at 96 kg N ha$^{-1}$ had potential to increase the growth, yield, nitrogen uptake, and NUE of maize by increasing the availability of N for plant growth and development.

**Keywords:** N losses; NBPT-coated urea (NCU); urease inhibitor; *Zea mays* var. Thai super sweet

## 1. Introduction

Granular urea is a nitrogen fertilizer commonly used in many countries, as well as in Malaysia's highly weathered tropical soils [1]. Urea is popular in developing countries due to its high N concentration (45–46% N), cheapness, easy handling, and lower production costs compared to other N fertilizers [2,3]. However, nitrogen loss via ammonia volatilization, denitrification, soil leaching, surface runoff, immobilization, or gaseous emission from urea granules under field conditions normally results in lower N uptake by plants as well as yield production [4]. The use of urea fertilizer has also been associated with relatively poor nitrogen use efficiency (NUE) due to these N losses [5,6]. In most crop fields, the overall efficiency of applied urea was lower than 50% [2]. The losses of N to the

atmosphere via ammonia volatilization and leaching from applying urea in most tropical soils were estimated at 30% to 60% [7] and 40% [8], respectively.

Types of fertilizers, climatic and soil conditions (soil pH, temperature, moisture, and wind velocity), agronomic practices, and management of fertilizer application are amongst the factors especially involved in affecting urea hydrolysis catalyzed by the urease enzyme [9]. Urease activity in soils under tropical conditions and highly weathered soil must therefore be controlled, for which urease inhibitors, such as N-(n butyl) thiophosphoric triamide (NBPT), are suggested as an effective way to inhibit urease activity and retard urea hydrolysis as well as reduce ammonia losses from urea [3,10–12].

Once applied to the soil, NBPT converts to active N-(n-Butyl) phosphoric triamide (NBPTO), which is the genuine inhibitor of urease activity. The bond angles and lengths of NBPTO are similar to urea, which enables the compounds to occupy the urease enzyme's active site to inactivate the enzyme and slow down urea hydrolysis [13]. Furthermore, NBPTO is not added into urea directly due to its instability, and it degrades faster than NBPT [3]. Many researchers proved that application of urea treated with NBPT has beneficial benefits compared to commercial urea fertilizer [14,15]. At present, the urease inhibitor NBPT, under the trade name of Agrotain®, has the potential to restrain urease activity, decrease N loss, increase NUE, and increase yield of crops [6,14,16,17]. This NBPT is easy to handle for treatment of urea, especially for farmers. The urea granules can be easily mixed with NBPT in a drum mixer and dried prior to application to soils.

However, the application rate of coated NBPT with urea for better plant growth is still globally debatable compared to normal practices (urea alone). Elsewhere, Khan et al. (2014) [11], in their study of arid calcareous soil in Pakistan, showed increasing grain yield and N uptake in maize in the plots fertilized with 115 kg N ha$^{-1}$ coated with 3 L ton$^{-1}$ of NBPT. On a similar note, as reported by Espindula et al. (2013) [18], the best N recovery by wheat plants was found when soils were applied with 100 kg N ha$^{-1}$ of urea + NBPT; however, they stated that the best NUE was found at the lower rate of 90 kg N ha$^{-1}$ of urea + NBPT.

Moreover, NBPT has been shown to contribute to an average increase of 0.89 t ha$^{-1}$ in maize productivity [15]. In addition, increases in herbage growth and N response efficiency of ryegrass with application of NBPT fertilizer were also reported by Dawar et al. (2012) [19]. In Malaysian sandy soil, research done by Mathiagan et al. (2019) [20] showed that NBPT had potential in decreasing the urease activity, hence slowing the rate of hydrolysis by less than 10% compared to urea. Delays in urea hydrolysis could be effective in lowering N losses as well as increasing N uptake and NUE [18]. Studies on urea coated with urease inhibitor and other materials, such as polymers and biochar, to slow the urea transformation were also conducted [21–24]. However, studies on the effects of the urease inhibitor NBPT when used to coat urea and when applied to soils at various rates on crops planted in highly weathered Malaysian soils is limited [20].

The efficiency of urea compared to NBPT-treated urea is still not understood. In maize cultivation, farmers normally applied 120 kg N ha$^{-1}$ in the form of urea. While the previous studies elsewhere showed that 90 to 115 kg N ha$^{-1}$ can be recommended, considering Malaysia's tropical and highly weathered soil, cultivation of maize with NBPT-treated urea and the effective application rate for enhanced N uptake and NUE are vital. Hence, this study was designed to test the hypothesis that NBPT-coated urea (NCU) could improve growth and yield of maize in tropical soils, with the following aims:

i.      To determine the effects of varying N rates of NBPT-coated urea (NCU) on the growth of maize.
ii.     To determine the best optimal NBPT-coated urea (NCU) application rate for the yield of maize.
iii.    To evaluate the effects of varying N rates of NBPT-coated urea (NCU) on nitrogen uptake and nitrogen use efficiency (NUE).

## 2. Materials and Methods

### 2.1. Geographical Location and Climatic Conditions

This field experiment was conducted at Field 15, Universiti Putra Malaysia, Serdang, Selangor, Malaysia, with the GPS coordinates of 2°59′04.9″ N, 101°44′01.7″ E, from October 2017 until January 2018. The selected field site had been under zero planting, with grasses as the surface cover or main vegetation. The average rainfall documented for the experimented region throughout the experimental period was 117.06 mm, with minimum and maximum temperatures of 21.9 and 38.0 °C, respectively (Climate Change Programme, Agrobiodiversity and Environment Research Centre, MARDI, Serdang).

### 2.2. Soil Condition

The soil samples were collected at 0 to 20 cm depth from the soil surface of undisturbed planting area for analysis prior to planting. The physico-chemical characteristics of the soil are tabulated in Table 1. The land was plowed with a rotovator, and Ground Magnessium Limestone (GML) liming of 2.5 t ha$^{-1}$ was applied to the soil to increase the soil pH. Soil pH before and after liming was analyzed using a Mettler Toledo pH meter with a ratio of 1:2.5 (soil to distilled water). Soils' total C and N were determined using the Dumas combustion method using a Carbon, Hydrogen, Nitrogen, Magnessium Laboratory Equipment Corporation (CHNS LECO) analyzer [25], available P was determined using the Bray No. 2 method [26], exchangeable K, Ca, Mg, and Cation Exchange Capacity (CEC) were determined using the leaching method [27], and soil mineral N ($NH_4$ and $NO_3$) analysis was determined using the steam distillation method [28]. The mechanical analysis of soil was done using the pipette method, and textural class was determined using the United States Department of Agriculture (USDA) soil textural triangle, and it was identified as *Typic Kandiudults* (Bungor series) and acidic.

**Table 1.** Physico-chemical characteristics of soil in the field.

| Parameter | Value |
|---|---|
| Soil textural class | Clay |
| pH before liming | 4.19 |
| pH after liming | 5.40 |
| Total (%) | |
| Sand | 40.79 |
| Silt | 14.29 |
| Clay | 44.92 |
| Total C (%) | 1.29 |
| Total N (%) | 0.18 |
| Available (mg kg$^{-1}$) | |
| P | 0.05 |
| K | 40.15 |
| Ca | 80.24 |
| Mg | 16.38 |
| CEC (cmol kg$^{-1}$) | 9.98 |
| Mineral N (µg g$^{-1}$) | |
| $NH_4^+$-N | 8.94 |
| $NO_3^-$-N | 6.52 |

### 2.3. Plant Material

The vegetation used for this experiment is maize (*Zea mays*) of the Thai Super Sweet variety, with planting distances of 25 × 75 cm. The size of each plot was specified to 5 × 5 m, and every plot had

6 beds with 20 plants in each bed. Direct sowing was done with 3 seeds per hole; after 9 days of planting, thinning (14 days) was done where only one plant remained per hole.

### 2.4. Preparation of NBPT-Coated Urea (NCU)

Coating urea with NBPT ensures that the NBPT is present only on the surface of the fertilizer granule. The NBPT-coated urea (NCU) was prepared in the Chemical Process Engineering Laboratory 2 at the Faculty of Engineering, Universiti Putra Malaysia (UPM) by spraying the NBPT at a standard volume (3 mL) on a 1 kg batch of urea, and then blending for 5 min at a constant speed (55 rpm) and angle (45°) in a SOLTEQ rotary drum, as recommended by KOCH Agronomic Services, Wichita, KS. This fertilizer was dried and kept in storage at room temperature prior to its application.

### 2.5. Experimental Design and Treatment Application

This experiment adopted Randomized Complete Block Design (RCBD) in which treatments were randomized in each block. The treatments for control were T0 (unfertilized), and T1 was the farmers' practice as recommended by *MARDI* (2010), in which NPK fertilizer ($15N:15P_2O_5:15K_2O$) was applied at an application rate of 400 kg ha$^{-1}$. The nitrogen fertilizer was applied 2 times (split application), which were during the growth period as basal fertilizer (2 weeks after planting) and as side dressing (5 weeks after planting) at 60 kg N ha$^{-1}$ for each application. Uncoated urea (46% N) was used as the N fertilizer source in T1, while for other treatments (T2–T5), NBPT-coated urea (NCU) was the source of N fertilizer with varying rates (144, 120, 96, and 72 kg N ha$^{-1}$). Treatment 2 (T2) was applied at a rate of 144 kg N ha$^{-1}$ of NCU (20% more than the T1 rate), T3 at a rate of 120 kg N ha$^{-1}$ (similar to the T1 rate), T4 at a rate of 96 kg N ha$^{-1}$ (20% less than the T1 rate), and T5 at a rate of 72 kg N ha$^{-1}$ (40% less than the T1 rate). For all treatments, all the fertilizers (N, P, and K) were applied at the soil surface. All treatments received similar frequency for P and K fertilizers and similar amounts of P (Triple Super Phosphate (TSP)) and K (Muriate of Potash (MOP)) ($60 P_2O_5$ and $60 K_2O$ kg ha$^{-1}$), including T0, which only did not receive any N fertilizer. The NBPT-coated urea (NCU) was provided by KOCH Agronomic Services, Wichita, KS. Details of treatments used in the experiment are shown in Table 2.

**Table 2.** Treatments used for the first cycle.

| Label | Specification |
|---|---|
| T0 | No Fertilizer |
| T1 | Farmers' practice (120 kg N ha$^{-1}$ urea) |
| T2 | NCU fertilizer (144 kg N ha$^{-1}$ with application frequency similar to T1) |
| T3 | NCU fertilizer (120 kg N ha$^{-1}$ with application frequency similar to T1) |
| T4 | NCU fertilizer (96 kg N ha$^{-1}$ with application frequency similar to T1) |
| T5 | NCU fertilizer (72 kg N ha$^{-1}$ with application frequency similar to T1) |

### 2.6. Data Collection and Analysis

At the end of the field experiment, which was 79 days after planting (DAP), two maize plants were randomly selected and harvested within a block per treatment (harvest time) as they reached the maturity stage. Leaf surface area (SLA) of the fresh leaves' surface area was determined using a Specific Leaf Area Meter while the leaves were still in a fully expanded and hardened shape. Grain yields were harvested from plants, excluding two border rows from each side in each treatment, and were determined from a net plot of 3 × 3 m. The cobs were removed, shelled, weighed, counted, and recorded. The samples were fully dried to identify moisture content (55–59%) and amount of water

before recording the dry weight. The yield of maize was expressed in t ha$^{-1}$ and was calculated as follows:

$$\text{Yield (t ha}^{-1}) = (\frac{Dry\ weight\ of\ yield\ (g)}{10^6}) \times (\frac{10^4}{9\ m^2}). \tag{1}$$

Nitrogen uptake was calculated by multiplying total plant N with plant dry weight [29], and nitrogen use efficiency (NUE) by determined as the percent of applied nitrogen that was consumed by plants. It was calculated using the difference method according to the following equation [30]:

$$\text{NUE (\%)} = \left[\frac{(\text{N uptake of fertilized plot} - \text{N uptake of unfertilized plot})}{\text{Rate of N applied}}\right] \times 100. \tag{2}$$

*2.7. Statistical Analysis*

All the collected data were analyzed using Statistical Analysis of System Software (SAS Version 9.4). ANOVA (Analysis of Variance) was used to determine the significant (a = 0.05) differences between the treatments, and Fisher's Protected Least Significant Difference (FPLSD) test was used to detect significant difference between means.

## 3. Results

*3.1. Effects of NCU on the Growth and Yield of Maize*

The growth and yield of maize in observations 79 DAP revealed that all NCU fertilizers at different rates had significant effects ($P \leq 0.05$) on chlorophyll content compared to the unfertilized plot (T0) and urea at 120 kg N ha$^{-1}$ (T1) (Table 3). Our findings also exhibited that application of NCU at a normal rate of 120 kg N ha$^{-1}$ and at lower rates (96 and 72 kg N ha$^{-1}$) (T3, T4, and T5) had similarly high overall plant dry weight compared to T0 and T1. Meanwhile, NCU applied at 120 kg N ha$^{-1}$ (T3) and NCU at 96 kg N ha$^{-1}$ (T4) were found to have significantly ($P \leq 0.05$) higher leaf surface area (SLA) compared to the controls (T0 and T1), respectively.

**Table 3.** Effects of N-(n-Butyl) Thiophosphoric Triamide (NBPT)-Coated Urea (NCU) treatments on chlorophyll content, plant dry weight, surface leaf area, and yield at 79 days after planting (DAP).

| Treatments | Chlorophyll Content (SPAD Unit) | Plant Dry Weight (g) | Surface Leaf Area (cm$^2$) | Yield (t ha$^{-1}$) |
|---|---|---|---|---|
| T0 | 37.71 b | 165.30 b | 1924.40 c | 1.86 d |
| T1 | 47.30 b | 234.15 b | 3123.20 bc | 3.80 c |
| T2 | 55.24 a | 288.82 ab | 3430.20 abc | 4.43 bc |
| T3 | 51.24 a | 403.05 a | 4928.70 a | 5.40 ab |
| T4 | 51.20 a | 359.86 a | 4543.90 ab | 5.65 a |
| T5 | 52.23 a | 375.22 a | 4537.80 ab | 4.72 abc |

(Note: T0; No fertilizer, T1; 120 kg N ha$^{-1}$ of urea, T2; 144 kg N ha$^{-1}$ of NCU, T3; 120 kg N ha$^{-1}$ of NCU, T4; 96 kg N ha$^{-1}$ of NCU, T5; 72 kg N ha$^{-1}$ of NCU. Means with different letters are significantly different at $P \leq 0.05$ according to Fisher's Protected Least Significant Difference (FPLSD) at 79 DAP).

In addition, NCU at a normal rate and lower rate (120 and 96 kg N ha$^{-1}$) (T3 and T4) significantly ($P \leq 0.05$) increased the yield of maize compared to the unfertilized plot and urea at 120 kg N ha$^{-1}$ (T0 and T1). The highest significant yield was obtained when 120, 96, and 72 kg N ha$^{-1}$ of NCU (T3, T4, and T5) were applied to soils, with mean yields of 5.4, 5.65, and 4.72 t ha$^{-1}$, respectively. The lowest yield (1.86 t ha$^{-1}$) was recorded in the plots where no fertilizer was applied. Notably, even the lower rate of NCU (T4) produced higher yield compared to the farmers' practice (T1). Therefore, almost 20% of fertilizer could be saved when using NCU as N fertilizer rather than urea to promote yield production.

### 3.2. Effects of NCU on Nitrogen Uptake and Nitrogen Use Efficiency (NUE)

According to Figure 1, maize plants grown in soils applied with 120 kg N ha$^{-1}$ of NCU (T3) and 96 kg N ha$^{-1}$ of NCU (T4) showed significantly ($P \leq 0.05$) and similarly greater N uptake as compared to the unfertilized plot (T0) and the plot with 120 kg N ha$^{-1}$ of urea (T1). This finding indicated that the T4 treatment, which was the lower rate of NCU (96 kg N ha$^{-1}$), is seen as the best rate and is recommended to increase N uptake. Therefore, about 20% of fertilizer usage could be saved when coating the urea with NBPT.

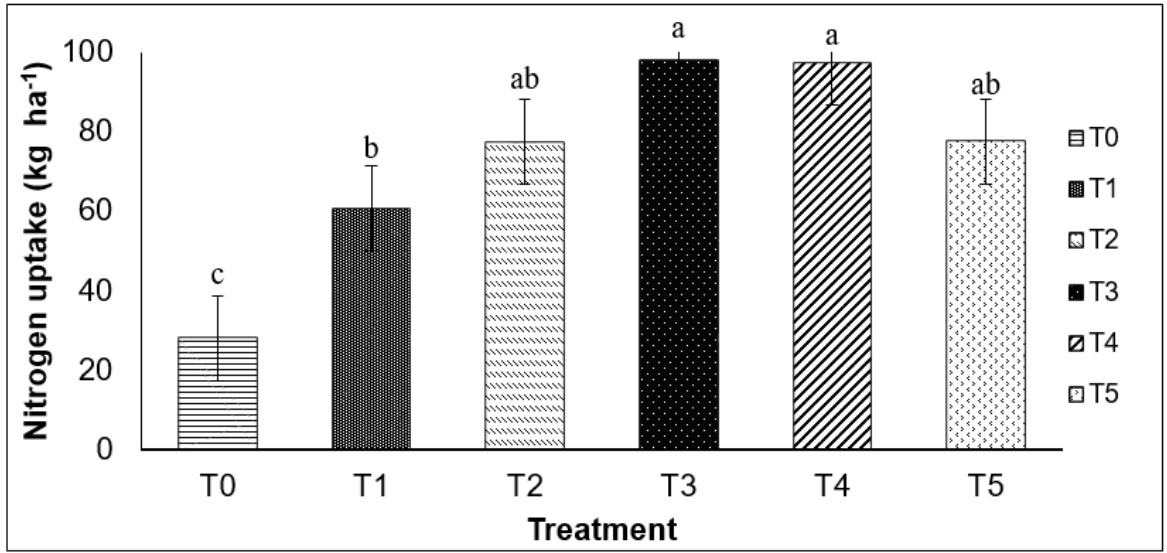

**Figure 1.** N uptake by maize plants at 79 DAP (Note: T0; No fertilizer, T1; 120 kg N ha$^{-1}$ of urea, T2; 120 kg N ha$^{-1}$ of NCU, T3; 144 kg N ha$^{-1}$ of NCU, T4; 96 kg N ha$^{-1}$ of NCU, T5; 72 kg N ha$^{-1}$ of NCU. Means with different letters on a bar are significantly different at $P \leq 0.05$ according to Fisher's Protected Least Significant Difference (FPLSD) at 79 DAP).

The nitrogen use efficiency (NUE) for various rates of NCU (T2, T3, T4, T5), the unfertilized plot (T0), and the farmers' practice (T1) on harvest time is illustrated in Figure 2. All NBPT-coated urea (NCU) fertilizer treatments had significantly ($P \leq 0.05$) higher NUE than that found with treatment of urea at 120 kg N ha$^{-1}$ (T1). The highest NUE (72.18%) was recorded for T4, where only 96 kg N ha$^{-1}$ of NCU was applied, and the lowest NUE (27.02%) was observed in the plot with only urea fertilizer (T1). Within the NBPT rate, T4 (96 kg N ha$^{-1}$ of NCU) and T5 (72 kg N ha$^{-1}$ of NCU), as well as T3 (120 kg N ha$^{-1}$ of NCU), gave similar NUE. Apart from that, the current results also showed that all NBPT treatments increase the NUE within the range 51.38%–72.18%. This implied that the applied nitrogen that was consumed by maize was within that range.

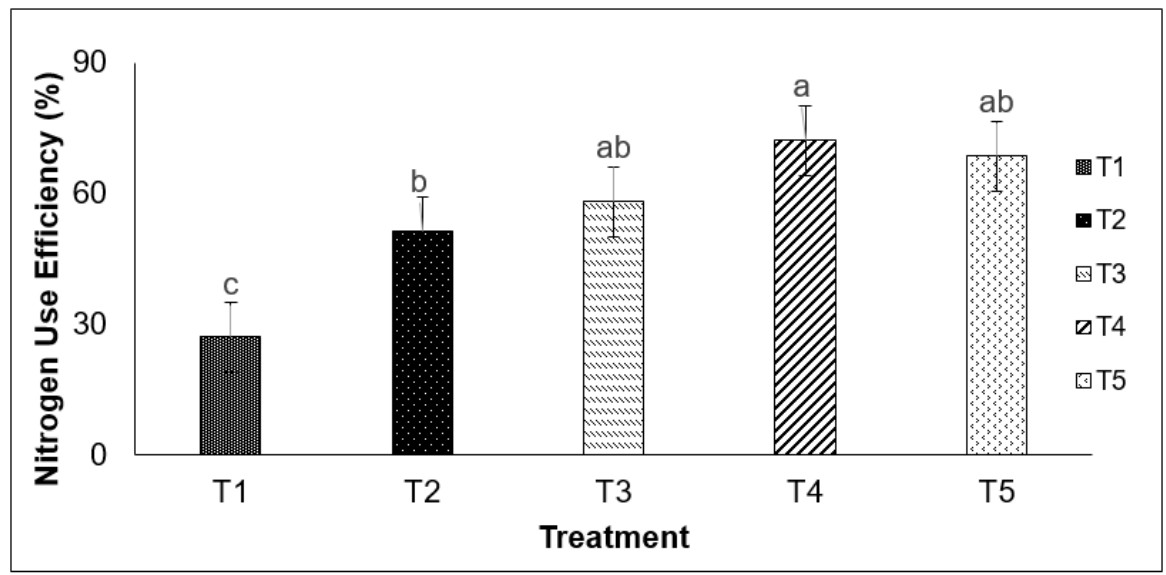

**Figure 2.** Nitrogen use efficiency (NUE) by maize plants at 79 DAP. (Note T1; 120 kg N ha$^{-1}$ of urea, T2; 120 kg N ha$^{-1}$ of NCU, T3; 144 kg N ha$^{-1}$ of NCU, T4; 96 kg N ha$^{-1}$ of NCU, T5; 72 kg N ha$^{-1}$ of NCU. Means with different letters on a bar are significantly different at $P \leq 0.05$ according to Fisher's Protected Least Significant Difference (FPLSD) at 79 DAP).

## 4. Discussion

### 4.1. Effects of NCU on the Growth and Yield of Maize

Nitrogen is often the most limiting nutrient for maize growth and yield, as it is a major constituent of protein and chlorophyll development [31]. However, rapid hydrolysis of urea has adverse effects on growth and yield of maize due to high N losses to the atmosphere in the form of ammonia ($NH_3$) [15], as well as via leaching [32]. In our study, coating urea with NBPT (NCU) showed significant effects on maize growth in terms of chlorophyll content, plant dry weight, and leaf surface area, and ultimately increased the yield of maize.

We observed greater effects of NBPT-coated urea (NCU) at 120, 96, and 72 kg N ha$^{-1}$ (T3, T4, and T5) in increasing plant dry weight compared to 120 kg N ha$^{-1}$ of urea (T1) and the unfertilized plot (T0). This phenomenon strongly suggested that NBPT-coated urea has potential to increase the dry weight of maize. The results of the significant increase in growth of maize due to the application of NBPT-coated urea (NCU) in this study were also supported by Dawar et al. (2011) [33], where the biomass yield of maize obtained with urea with NBPT was significantly higher than that obtained without NBPT. Khan et al. (2014) [11] also reported the improvement in yield parameters like number of rows of ears, number of grain rows, and number of grain ears due to the application of NBPT.

To support our indication, it can be seen that NBPT-coated urea at 120 kg N ha$^{-1}$ (T3) resulted in significantly ($P \leq 0.05$) higher SLA compared to 120 kg N ha$^{-1}$ of urea (T1). The results revealed the function of NBPT in delayed the hydrolysis, which would make N stay longer in soil and help the roots absorb the N sufficiently, therefore positively contributing to plant vegetative growth and to the expansion and elongation of the plant leaves [18].

This current results also showed that NCU fertilizer at rates of 120 and 96 kg N ha$^{-1}$ significantly increased the yield of maize compared to the unfertilized plot and urea at the recommended rate (120 kg N ha$^{-1}$). Corroborating these results, Espindul et al. (2013) [18] reported in their research that NBPT at 90 kg N ha$^{-1}$ increased the yield of wheat variety "BRS 254" more than 4% as compared to urea at 120 kg N ha$^{-1}$. At similar rates of fertilization with 120 kg N ha$^{-1}$ of NBPT or urea, NBPT positively increases yield more than 13% [18]. This increase in yield of maize by more than 21% due to

the application of NBPT as compared to urea alone was also recorded by Venterea et al. (2011) [34] in their study in 2009.

This increase in growth (chlorophyll content, plant dry weight, and leaf surface area) and yield from the results in this current study could be due to the function of the urease inhibitor NBPT in delaying the hydrolysis by disrupting the activity of the urease enzyme in soil, which resulted in delaying the rate of urea hydrolysis by up to two weeks [13,15]. The three active sites of the urease enzymes were intensely blocked by NBPT, causing it to form a bond of a tridentate nature, with two nickel centers and one oxygen, which resulted in higher efficiency in delaying urea hydrolysis [3]. This delay in hydrolysis would lower $NH_3$ losses and increase the availability of N for plant uptake, which resulted in indirectly increasing the performance and yield of maize. As stated by Upadhyay (2012) [35], it protects soil from losing nitrogen after the use of urea fertilizer by controlling hydrolysis of urea in soil. This phenomenon would reduce the $NH_3$ losses and enable plants to take more available N through their roots [3,10]. Reduction of N losses due to the application of NBPT and, ultimately, increases in the plant growth and yield were also reported by Zaman et al. (2010) [9] and Chien et al. (2009) [14].

Significant results of urease inhibitor (NBPT) on other crops' growth and yield were also reported by other scientists. Li Min et al. (2010) [16] reported increase in grape yield by 22.6–27.82% when applying NBPT. Similar results were found by [36,37], where the addition of urease inhibitor (NBPT) significantly increased dry matter yield and the grain weight of rice. These findings can also be supported by McClallen (2014) [38], who found a significant impact on wheat yield production by applying NBPT fertilizers.

In terms of the application amount of NCU, the results obtained in this study showed considerable potential of NCU for improving the dry weight of maize and yield at lower rates of NBPT-coated urea (96 kg N ha$^{-1}$), instead of at similar corresponding rates of urea (120 kg N ha$^{-1}$). Almost 20% of nitrogen fertilizer could be saved, leading to greater economic return. Similar findings were reported in the research done by Zaman et al. (2010) [9] and Karamanos et al. (2004) [39], where an increase in wheat grain yield was noted when NBPT was applied at a lower rate and similar results were produced with as low as one third of the recommended rate, respectively.

## 4.2. Effects of NCU on Nitrogen Uptake and Nitrogen Use Efficiency (NUE)

The positive effects of NBPT-coated urea (NCU) at 96 kg N ha$^{-1}$ showed the inhibitory effect that NBPT has on the rate of urea hydrolysis. The NBPT, which acts as a urease inhibitor, was able to limit urea hydrolysis, which was then reflected by the N uptake and higher NUE. Similar to these results, Zaman et al. (2013) [40] and Watson (2000) [41] reported that urea applied with NBPT had higher N uptake than the urea treatment alone. NBPT-treated fertilizer acted as a slow release fertilizer, resulting in maximum rice N uptake [36]. Our findings were in support of those of Sanz-Cobena et al. (2008) [42] and Goos (2011) [43], who found significant reduction in N loss when plants were treated with NBPT fertilizer. The poor N uptake associated with urea treatments alone could be due to the large volatilization [23]. NBPT has the capability to delay the urea hydrolysis and control N losses to air and ground, along with improvement in N uptake and NUE. Numerous scientists have done meta-analysis on NBPT and showed positive results in reducing N loss and increasing NUE as well as crop yield [3,17,44]. In addition, our findings were also in line with those of Kawakami et al. (2012) [45], where it was reported that addition of NBPT increased N uptake by 17% and N use efficiency by 41%. Higher NUE showed that higher available N consumed by plants will promote the plant growth and increase the crop yield. Espindula et al. (2013) [18] also found that NBPT was able to improved NUE in rice by 16.52%.

Moreover, the lower amount of NCU fertilizer (96 kg N ha$^{-1}$) application resulted in higher NUE, and it was expected that it would increase the economic return (which was not in this study's scope). Reference [15] also reported that the agronomic efficiency was higher when NBPT was applied at a lower N level (180 kg N ha$^{-1}$). This study suggested that, rather than using the normal rate of NCU

($120$ kg N ha$^{-1}$), the lower rate of NCU ($96$ kg N ha$^{-1}$) had the potential to perform better in increasing the NUE. Reference [16] verified that the N use efficiency and N uptake by plants reached their highest when the amount of NBPT added was only 0.5% of the total urea N applied.

## 5. Conclusions

This research on the effects of varying rates of NBPT-coated urea (NCU) versus uncoated urea on maize indicated that a lower application rate of NBPT-coated urea ($96$ kg N ha$^{-1}$) (T4) was the most suitable fertilization rate; it is therefore recommended for surface application in tropical soils for maize cultivation. The urea coated with NBPT has been shown to slow down the hydrolysis process of urease decomposition and to reduce NH$_3$ loss. Thus, it has the potential to improve the efficiency of urea by increasing chlorophyll content, plant dry weight, crop yield, and nitrogen use efficiency (NUE). Instead of using the normal rate of NCU ($120$ kg N ha$^{-1}$) or $120$ kg N ha$^{-1}$ of urea (farmers' practice), the lower rate of NCU application was efficient in delaying the hydrolysis of urea, which resulted in increasing plant bioavailability. However, the results of our study were limited to maize planted for one season; hence, more field work is needed to determine the effects of NBPT in highly weathered soil.

**Author Contributions:** Conceptualization M.M.M.Z.; methodology, M.M.M.Z. and M.K.Y.; formal analysis, M.M.M.Z., N.M.J. and S.Z.S.; investigation, M.M.M.Z. and M.K.Y.; resources, M.M.M.Z.; data curation, M.M.M.Z., N.M.J., S.Z.S. and M.K.Y.; writing-original draft preparation, M.M.M.Z.; writing-review and editing, M.M.M.Z., N.M.J. and M.K.Y.; visualization, M.M.M.Z.; supervision, M.M.M.Z. and N.M.J.; funding acquisition, N.M.J. All authors have read and agreed to the published version of the manuscript.

**Funding:** The research received funding from Koch Fertilizer International Limited from 2017 to 2019.

**Acknowledgments:** The authors are thankful to KOCH Industries, Inc. for supplying NBPT-coated urea (NCU) fertilizers and helping with financial needs, as well as Universiti Putra Malaysia for providing the facilities and field area to conduct this experiment.

**Conflicts of Interest:** The authors declare no conflict of interest.

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
