# Peer review of "N-(n-Butyl) Thiophosphoric Triamide (NBPT)-Coated Urea (NCU) Improved Maize Growth and Nitrogen Use Efficiency (NUE) in Highly Weathered Tropical Soil"

_sustainability, doi:10.3390/su12218780_

Round 1

Reviewer 1 Report

The paper deals with the effect of use urea coated with a urease inhibitor on the yield and NUE of maize crop. The paper has valuable and laborious data in order to understand those effects. Overall the manuscript is well done. The discussion, the presentation of the results and the conclusions are consistent with the data obtained during the experiment and are also appropriate for the audience. 

More please see attached.

Author Response

Dear reviewer, 

Thank you for your comments. Really appreciate it. 

Reviewer 2 Report

The manuscript "N-(n-Butyl) Thiophosphoric Triamide (NBPT) coated urea (NCU) improved maize growth and nitrogen use efficiency (NUE) in highly weathered tropical soil" is well written- short and concise. It is well written and will be highly suitable work to Malaysia and similar regions. I do not have further comments to improve the quality of manuscript.

[Suggestion] change title in other way than presenting results

[Suggestion] At the beginning of sentence, please write full form. For example, change ‘N’ into ‘Nitrogen’ in abstract.

[Suggestion] include number of growing days in Line #109

Good luck!

Reviewer 3 Report

Overall:

Important exploration of an intervention that can have economic and environmental benefits.

In this paper, there are some claims that NBPT treatments at a given level of nitrogen application improved NUE, crop growth, or yield relative to the control. However, there are not a suite of controls wherein uncoated urea was applied at the same rates that NBPT urea was applied. Therefore, this is not a valid comparison, because you have not controlled the variables. You can compare T3 to T1, and say that NBPT improved parameters as compared to T1, and attribute those changes to the NBPT coating. You can also say that a certain rate of NBPT coated urea is the Agronomically Optimal Nitrogen Rate (AONR) for maize at your research station. Your paper currently doesn’t compare T2 to T1, which would be problematic. I can accept a comparison such as lines 167-169, in which you state that lower rates of NCU perform better than recommended uncoated urea rates. Please take care to clarify when you are comparing different nitrogen rates, for example in the abstract, and contextualize that comparison.

Also, instead of referring to treatments as normal rate and lower rate, simply refer to them by the amount of nitrogen that you applied. I do like how frequently you define T1, T2, etc throughout the paper.

It sounds like from your introduction that you are interested in calculating the AONR for NCU in your setting, however, you haven’t presented this in your results. Consider adding this regression analysis.

Ln 16- change solution to solutions

Ln 16- change reduces to reducing

Ln 23- the control fertilized at the same level with uncoated urea?

Ln 24- compared to control at similar fertilization levels?

Ln 26- insert comma between yield and nitrogen

Ln 31- replace Malaysia with Malaysia’s

Ln 34- remove extra space

Ln 36- replace condition with conditions

Ln 50- replace inactivating with inactivate

Ln 59- state author’s name- in general do this throughout when you are using in-line citations. For example “as discovered by Jones et al. [13], …” or “Smith et al. (1989) found that…”

Ln 58-63- is this paragraph about urea rates or NBPT rates or both? Please make the topic sentence congruent with the content of the paragraph

Ln 65- replace increased with increases

Ln 65- replace when with with

Ln 68- replace delayed with delays

Ln 68- replace affecting with effective

Ln 74- replace this sentence with “The efficiency of urea compared to NBPT treated urea is still not understood”

Ln 95- replace soils with soil

Ln 100- replace soils with soil

Ln 105- remove extra space

Ln 121- replace “Completely… Design” with Randomized Complete Block Design

Ln 136- replace “similar as” with “similar to”

Ln 138- replace day with days

Ln 143- did you shell the corn before weighing to determine grain yield? Cobs have weight….

Ln 138-144- did you dry down the maize fully and adjust for target moisture content? At what percent moisture was your maize when you reported the yield?

Ln 145-147- your formula and text disagree. Your formula is in line with other NUE calculations I’ve seen.

Ln 151- which LSD test did you use?

Ln 157- replace finding with findings

Ln 160- define SLA

Ln 180- confusing- which treatment had greatest N uptake? Why would you recommend one if another was better?

Ln 227- replace increase with increases

Ln 228- remove of before more

Ln 228- replace increased with increase

Ln 234- replace block with blocked

Ln 235- replace delayed with delay

Ln 237- you have a citation in a different format here. Do state the author’s name, as indicated before, but also assign a number to the citation

Ln 240- replace reducing with reduced

Ln 242- remove comma

Ln 244- replace result with results; insert were before found

Ln 248- remove is

Ln 261- replace plant with plants

Ln 270- replace greater with increase

Ln 273- replace performed with perform

Ln 285- you didn’t measure N losses, so you don’t know for certain this happened- reword to indicate that this is speculation
